# Brief communication: On the potential of seismic polarity reversal to identify a thin low-velocity layer above a high-velocity layer in ice-rich rock glaciers

Jacopo Boaga [(1),] Mirko Pavoni [(1)], Alexander Bast [(2)(3)], Samuel Weber [(2)(3)]

[(1)] *Department of Geosciences, University of Padova, Via Gradenigo 6, 35131 Padova, Italy*

[(2)] *WSL Institute for Snow and Avalanche Research SLF, Flüelastrasse 11, 7260 Davos Dorf, Switzerland*

[(3)] *Climate Change, Extremes and Natural Hazards in Alpine Regions Research Center CERC, Flüelastrasse 11, 7260 Davos Dorf, Switzerland*

*Correspondence to*: Jacopo Boaga (jacopo.boaga@unipd.it)

**Abstract.** Seismic refraction tomography is a commonly used technique to characterize rock glaciers, as the boundary between unfrozen and ice-bearing layers represents a strong impedance contrast. In several rock glaciers, we observed a reversed
polarity of the waves refracted by an extended ice-bearing layer compared to direct wave arrivals. This phase change may be related to the presence of a thin low-velocity layer, such as fine- to coarse-grained sediments, above a thicker ice-rich layer. Our results are confirmed by modelling and analysis of synthetic seismograms to demonstrate that the presence of a low-velocity layer can produce a polarity reversal on the seismic gather.

## 1. Introduction

Rock glaciers are prominent landforms in permafrost environments that pose a potential mass movement hazard, endangering communities and infrastructure in high mountain regions. As their kinematics are strongly dependent on the presence of ice and water, various geophysical methods have been used in recent decades to map, characterize, and monitor the internal structure of these permafrost-related landforms. Seismic methods are one of the most frequently and earliest applied methods
to investigate the near subsurface of rock glaciers besides electrical methods such as electrical resistivity tomography (ERT; e.g., Hilbich et al., 2010), induced polarization (e.g., Duvillard et al., 2018) or electro-magnetic methods (e.g., Boaga et al., 2020; Pavoni et al., 2023b). Recently, refraction seismic tomography (SRT; e.g., Musil et al., 2002) has regained popularity, as joint-inversion algorithms provide a more detailed insight into the ice, water, rock and air composition within a rock glacier (e.g., Hauck et al., 2011; Pavoni et al., 2023a). SRT is a very suitable method because the refracted seismic wave travels along
the ice-bearing layer at higher seismic velocities than the upper sediments. Typically, higher velocities at depth imply higher impedance of acoustic contrast ($Z$), which is the product of density ($\sigma$) and seismic velocity ($v$). Reflection coefficient R influences the energy reflected at the boundary:

$$R = (Z_2 - Z_1)/(Z_2 + Z_1) \qquad\qquad (1)$$

where $Z_1$ is the acoustic impedance of the overlying media, and $Z_2$ is the acoustic impedance of the bottom one. The boundary
generates critically refracted waves if the reflection coefficient (R) is positive (Harvey et al., 2011). The seismic velocity structure of the ground can be retrieved from the first arrival times of the direct and refracted waves (Leopold et al., 2011). Since critically refracted waves are generated in the case of a positive reflection coefficient ($R$), a possible low-velocity layer (LVL) between two faster ones is not visible using the seismic refraction technique, which is the method's main drawback.

However, different wave attributes can be analysed in the seismic shot gather. A polarity reversal, or so-called phase change, is an example of a local amplitude seismic attribute anomaly in reflected waves that can indicate the presence of LVL between two faster media. For example, phase polarity inversion is considered a direct hydrocarbon indicator in seismic hydrocarbon exploration. Polarity inversion has been observed in several glaciological seismic reflection studies (e.g., Anandakrishnan et al., 2003). The phase shift can also be observed in the vertical component of critically refracted head-waves, in this case due to the complex interaction of the wave with a boundary or a change in the medium, as a phase lag generated by slow layers presence. We observed such a reversal of the polarity of the refracted waves, with respect to the direct wave phase, in SRT datasets of several rock glaciers. Here, we present two case studies from Switzerland, where two rock glaciers were characterized using ERT and SRT, and in one case with additional borehole information. In both cases, the refracted wave recorded with vertical geophones shows a reversal polarity in correspondence with the buried ice-rich layer. We speculate that this phase change may be due to the presence of an LVL, such as fine- to coarse-grained sediments with ice, overlying a high-velocity layer of massive ice, or with ice-filled veins between coarse blocks (see Figs. 1c and 2c). Such a LVL in RGs is typically a leach product of upper blocks, consisting of fine sediment of variable thickness (often < 1 m), with moderate resistivity (< 2 KOhm m) and poor mechanical properties (Vp < 500 m/s). This LVL is hard to be detected by ERT, if the thickness is less than the electrode spacing, and is apparently undetectable by SRT.We computed synthetic seismograms for two subsurface models, with and without the LVL. The synthetic modelling confirms that in our case study, the presence of even a thin layer of fine sediment with low seismic velocity can induce a reversal in the polarity of the refracted waves.

## 2. Real data case studies: sites, methods, and results

In this study, we investigate the Schafberg and Flüelapass rock glaciers (RGs), Canton Grisons, Eastern Swiss Alps. At both sites, ERT and SRT data were collected in August 2022. The ice-rich Schafberg rock glacier is located above Pontresina at 2750 m a.s.l. (~ 46.49 N, ~ 9.93 E). We performed the measurements nearby four boreholes. One borehole was drilled in1990 and equipped with thermistors (Vonder Mühll and Holub, 1992). In 2020, three further boreholes were drilled and equipped with piezometers, temperature sensors, and a cross-borehole ERT setup (Phillips et al., 2023, Bast et al. 2024). The stratigraphies recorded during drilling indicate a 3 – 4 m thick layer of boulders above a layer of fines with ice (~ 1 m), over coarse sediments with ice, and a layer of ice and/or mud with ice (Phillips et al., 2023) and found bedrock around 16m depth. The internal structure was confirmed by ERT, SRT, and electromagnetic soundings (Boaga et al., 2020, Pavoni et al., 2023a). ERT and SRT acquisition was performed with 48 channels and 3 m spacing between the electrodes, or geophones, respectively. The Flüelapass rock glacier is an active rock glacier complex located at the pass between the Flüela and Susasca Valleys in the Eastern Swiss Alps (~ 46.75 N, ~ 9.95 E). This RG is a creeping ice-rich landform that ranges from 2380 to 2800 m asl., surrounded by steep rock walls consisting mainly of amphibolite and paragneiss. The rock glacier surface is made up of debris and boulders of various sizes. Here, we conducted ERT and SRT soundings with 48 electrodes and geophones having 2 m spacing. The inverted tomograms suggested the presence of an ice-bearing layer beneath a 3-5 m thick layer of unfrozen debris in the upper part of the section. Data were collected and processed in the same way at both sites. For data acquisition, we used

a Syscal Pro Switch 48 georesistivimeter (*Iris Instruments, www.iris-instruments.com*) with a multi-skip acquisition scheme and direct and reciprocal measurements to define a reliable expected data error (10% in Fluelapass and 20% in Schafberg).

Multi-skip dipole acquisition ensured good depth sensitivity and lateral resolution (Pavoni et al., 2023a), however the spacing between the electrodes still determines the thickness of the observable layer, making any thin LVL invisible. Seismic data were recorded with Geode seismographs (*Geometrics, San Jose, USA, www.geometrics.com*) using vertical geophones (100 Hz) for both sites. Shot locations were at every second geophone, and signals were triggered with a sledgehammer (20 kg). Sampling was 0.25 *ms* and the signal-to-noise ratio was improved by stacking the traces recorded twice at each

location. In this way, we facilitated the manual picking procedure of the compressional wave's first-time arrivals and assessed uncertainty (picking error = 2 *ms*) by performing repeated picking (Pavoni et al., 2023a). Inversion modelling was performed adopting unstructured triangular meshes to accurately consider the irregular topography of the rock glacier environments (we measured at each electrode/geophone with a Stonex S800 GNSS receiver, *www.stonex.it*). ERT inversions were performed using the open source Python-based software ResIPy (Blanchy et al., 2020) applying a weighted least squares objective

function with normal isotropic regularization inversion and linear filtering. Both the models reached the convergence criteria (final RMS misfit<1) in 2 iterations. For SRT inversion modelling we adopted the open-source C++/Python-based library Pygimli (Rucker et al., 2017). In this case, the optimal regularization factor was defined with a L-curve analysis, and both the Vp models reached the convergence in 4 iterations. In the raw seismograms of both sites, we observed a reversal polarity of the refracted waves' first arrivals with respect to the direct waves' first arrivals, corresponding to the presence of an ice-bearing

layer (Figs. 1 and 2). The appendant figure panels *a* and *b* show the results of the inversion processes of the SRT and ERT datasets. The inverted models reveal the internal structure of the rock glaciers, and the estimated ice-bearing layer boundary (dashed lines), estimated by applying the steepest gradient method to the ERT results (Pavoni et al., 2023c). Panels *c* show the interpreted models, as confirmed by boreholes information in the Schafberg case. Panels *d, e* and *f* show the raw seismograms for left, central and right shots, respectively. Wiggle seismograms mode is shown with red (+) and blue (-) phase colours of

the shot gather, to highlight the polarity reversal.

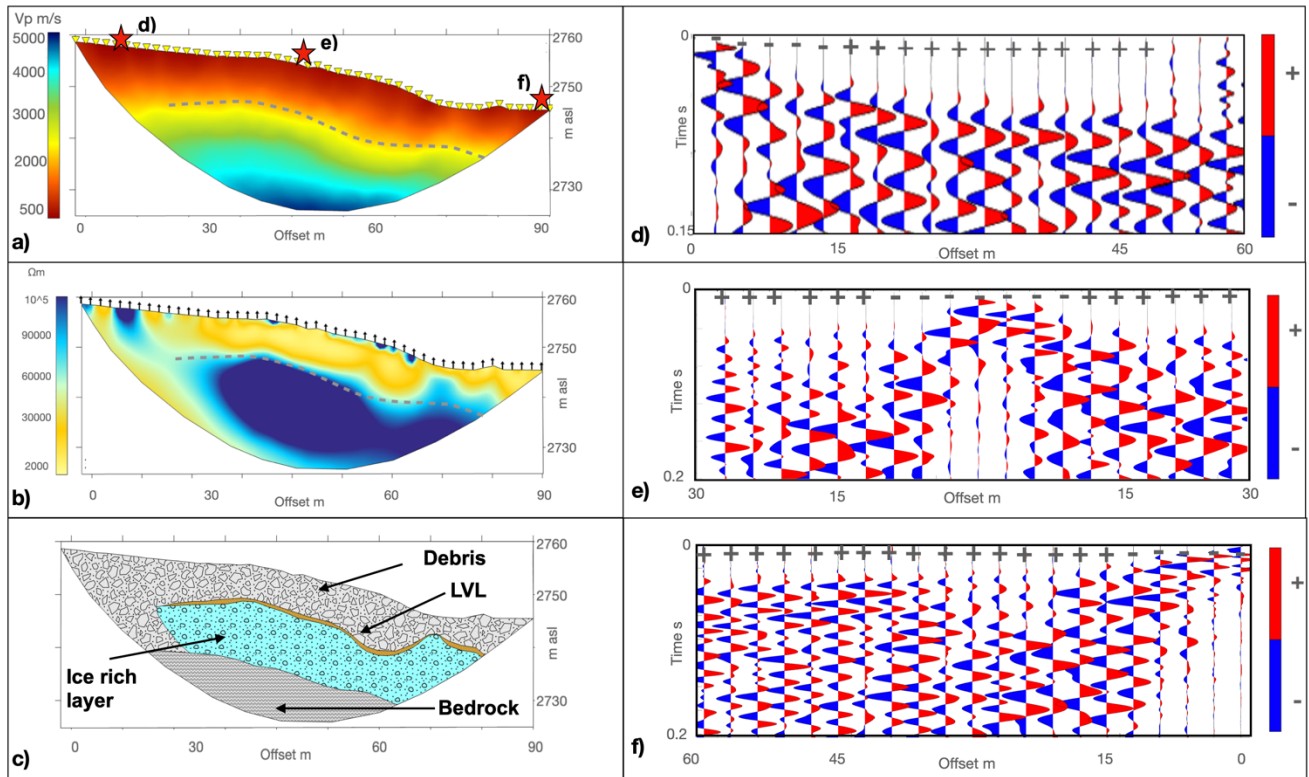

**Figure 1: a)** Schafberg SRT inverted section red stars are the shot positions here presented. **b)** Schafberg ERT inverted section. The dashed line represents the ice-bearing layer boundary evaluated with the steepest gradient method applied to the resistivity section. **c)** Schafberg interpreted model based on ERT and SRT profiles, and boreholes stratigraphies (Phillips et al . 2023). Panels d), e), and f) show the wiggle mode (zoom) seismograms above the ice-bearing layer for the lateral and central shots. The seismograms highlight the reversal polarity of the refracted waves: first direct wave arrivals have blue-negative polarity (-), while refracted wave first-arrivals switch to red positive polarity (+).

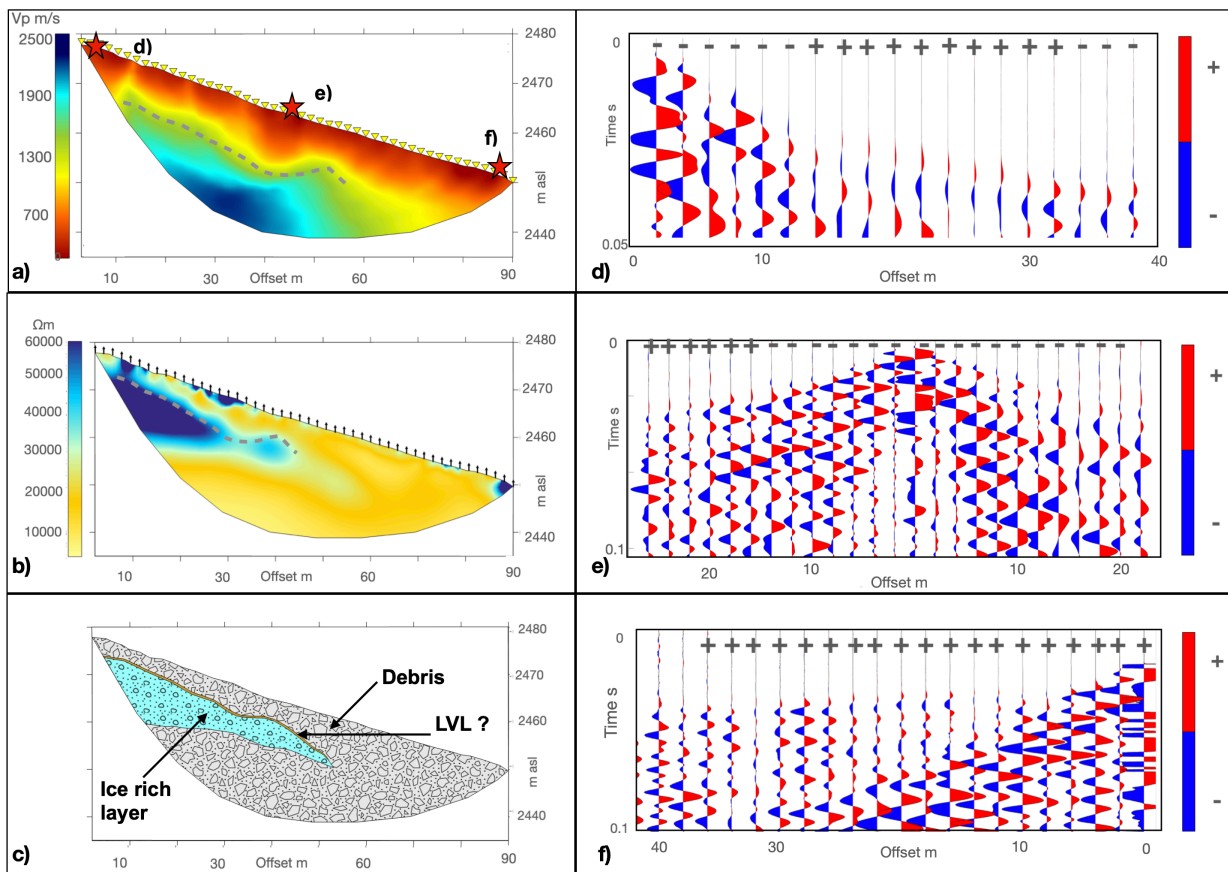

**Figure 2: a) Flüelapass SRT inverted section , red stars are the shot positions here presented; b) ERT inverted section. The dashed line represents the ice-bearing layer boundary evaluated with the steepest gradient method applied to the resistivity section; c) Flüelapass interpreted model based on ERT and SRT results. d), e), f) are the wiggle modes (zoom) seismograms above the ice-bearing layer for lateral and central shots. The seismograms show the reversal polarity of the refracted waves, but only in the upper section where an ice-rich layer is present (d, e,). Lateral shot (f) does not exhibit polarity reversal.**

At both sites, the shot gathers show an evident phase change from negative to positive between the direct waves' first arrivals (blue, negative polarity) and the critically refracted waves' first arrivals (red, positive polarity) in the presence of the ice-bearing layer boundary. As it can be seen in Figure 2c, in the Flüelapass case, the ice-rich layer and the possible LVL are limited to the upper part of the RG; accordingly, the central and right shots (Fig. 3, panels *e* and *f*) do not exhibit phase reversal in the lower part. We hypothesize that the observed reversal polarity can be attributed to the interference generated by the presence of a thin LVL of fine- to coarse-grained sediments with ice and unfrozen water above a high-velocity ice-rich layer, as documented by the stratigraphy of the Schafberg rock glacier (Phillips et al., 2023).

### 3. Representation in a synthetic model

We computed synthetic seismograms based on information from the Schafberg boreholes, SRT and ERT data. The forward problem was implemented using the advanced full-waveform spectral element solver Salvus (by the Mondaic ETH spin-off; see Afanasiev et al., 2019). We use a ricker wavelet source centered at 50Hz to correctly simulate our sledgehammer shot (having power frequency in the range 10-90 Hz), adopting an adaptive mesh of 10 elements per wavelength and absorbing external boundaries with100 receivers spaced 1 m apart (see Salvus documentation in Afanasiev et al., 2019). Figure 3 shows the results of the schematic layered model for the Schafberg structure, in one case without LVL within the structure (Fig. 3, panels *a* and *c*), and then adding a 1 m thick LVL having Vp= 400m/s, Vs= 220 m/s, δ =1.6 g/cm$^3$ (Fig. 3, panels *b* and *d*). For both the models, the other parameters are: upper debris Vp = 600 m/s, Vs= 400 m/s, δ =1.9 g/cm$^3$; ice rich layer Vp= 3000 m/s, Vs = 1600 m/s δ =2.0 g/cm$^3$; bottom layer Vp = 4000 m/s, Vs = 2200 m/s, δ =2.3 g/cm$^3$.

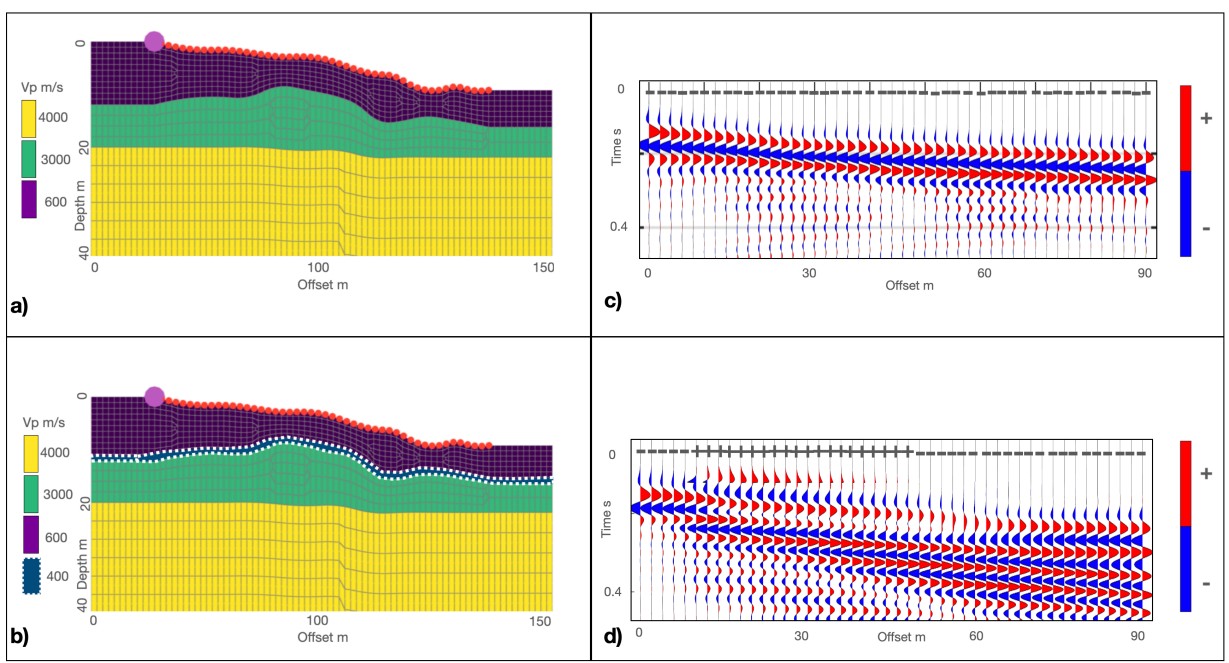

**Figure 3. Synthetic models of the Schafberg rock glacier a) without and b) with the fine sediments 1 m thick LVL (blue zone between dashed white lines). The purple dot is the source position, and the red dots are the receivers. c) Synthetic seismograms computed without the LVL presence; d) synthetic seismograms computed with the same model but with the presence of 1 m thick LVL.**

### 4. Discussion and Conclusions

The presence of a thin LVL consisting of fine sediments significantly increases the complexity of the shot gather (see Fig. 3, panels *c* and *d*). The synthetic seismograms confirmed the field results: the seismogram with the LVL layer shows a phase change that is not present in the seismogram without LVL. In the synthetic model with the LVL, the polarity of the first arrivals of the refracted waves is in fact reversed compared to the phases of the direct waves (from the negative phase in blue to the positive phase in red). On the contrary, in the absence of an LVL, the direct and refracted waves retain the same polarity

without phase change (see Fig. 3c). Furthermore, the synthetic shot gather in Figure 3d is very similar to the actual shot gather collected at Schafberg (Fig. 1d), where borehole stratigraphy confirmed the presence of a thin fine-sediment layer above a high-velocity ice-rich layer (Phillips et al., 2023). This observation suggests that the observed polarity reversal at Schafberg and Flüelapass (also observed at other sites not presented here for brevity) may be due to the presence of an LVL of finer sediments accumulating on top of the ice-rich layers. This thin LVL is difficult to detect by geophysical imaging because conventional ERT does not have the necessary resolution and SRT is incompatible with velocity inversion at depth. This thin LVL may play a relevant role in the hydrological behavior of supra-permafrost water fluxes in rock glaciers (Jones et al., 2019), most likely composed of low-permeability fine sediments. This LVL may, for example, help the ice-rich layer to act as an aquiclude (Pavoni et al., 2023c) or favour local water accumulation (Haeberli et al., 2001). Therefore, the simple observation of phase reversal in the shot-gather of the seismic refraction dataset can be interpreted as a proxy for complex subsurface structure, and this cannot exclude the presence of fine-sediments overlying the ice-rich/massive ice zone. Once the possible presence of an LVL has been suggested, future perspectives may include its characterization, both in terms of thickness and continuity, by specifically designed surveys as very high-resolution ERT (i.e. with very little electrodes spacing) or detailed surface waves analyses (Barone et al., 2021).

*Author contributing.* JB developed the concept of the study, AB, JB, SW, MP have been involved in data acquisition; MP and JB performed the data processing; all authors contributed to the writing and editing of the manuscript.

*Data availability.* The datasets used to obtain the results and all the processing technical info will be provided by authors on recquest.

*Competing interests.* The authors declare that they have no conflict of interest.

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
