# Peer review of "Brief communication: On the potential of seismic polarity reversal to detect a thin low-velocity layer above a high-velocity layer in ice-rich rock glaciers"

_EGUsphere, 2023_

## Author Response (AR1)

**EGUSPHERE-2023-2774-R2**

**Reply to Editor**

*Dear Editor,*
*We present here a response to your comment (your comments in black, our responses in italic red).*
*Jacopo Boaga (on behalf of all authors)*

Dear authors
thank you very much for submitting your manuscript to this special issue. The two reviewers provide some very detailed comments on your manuscript, which I invite you to address. I very much agree with their comments in that this is an interesting paper (wich addresses phase reversals that I have seen in my own data), but you may want to provide some additional details on your modelling, and perhaps a changed modelling approach to be clearer on the origin of the phase reversal and whether this can be classified as a refracted wave or not. In particular, I was also wondering, whether perhaps a simpler model, that does contain less complexity to highlight the occurrence of the phase reversal and/or adding some ray paths to the Fig. 3 would provide further insights the dynamics taking place here. For Fig 1 and 2, it would be good to add the shotpoint and the used geophone locations for the shown shot gathers, as your spread appears to not be covering the entire profile, and hence it is difficult to understand along which portion of the profile this is actually occurring.
One very minor comment, please double check your references; Binley 2015 is cited but appears to be missing in the reference list.
Sebastian Uhlemann

*Dear Editor,*
*We have carefully addressed all the reviewers' comments. Specifically, in the response to Rev1, we included a simple flat layer to show that topography isn't responsible for the observed phenomena (see fig.1 of this reply). We fully agree with Rev1 and have avoided any simplistic statement about the presence of an LVL and the observed polarity reversal, as the polarity of the head waves should not change. We are in fact observing, as Prof. Maurer correctly pointed out, a complex wave interference due to the decrease in velocity in the LVL. On the other hand, since the layers thicknesses are confirmed by other independent measurements such as ERT and moreover by boreholes, we believe in the correctness of our time picking.*
*The manuscript's figures have been completely redrawn based on your and Rev2 comments. Shot points and receivers have been added, along with multiple shots in wiggle mode view. Since we have a maximum of 3 figures in the 'brief communication', we prefer to avoid additional panels that make the graphics difficult to read. That's why we prefer to*

*add the flat layer figures in the reply to rev1 and not add features to fig.3, as this discussion is public and open to all interested readers of TC. The reference list is now correct, we cannot exceed 20 references, but all the ERT processing specifics can be found in the cited papers. We still believe that this is an important message to convey to the cryosphere community to avoid a simplistic interpretation of the subsurface in the common use of reflection seismic tomography in rock glaciers. Thank you for your time.*

**Reply to Rewier1**

*Dear Prof. Maurer,*
*Thank you for your time. We present here a response to your comment (your comments in black, our responses in italic red).*
*Jacopo Boaga (on behalf of all authors)*

The paper by Broaga et al. presents results of a seismic refrac7on tomography (SRT) study on a rock glacier. They infer that a low-velocity layer (LVL) can be detected by means of polarity reversals of the first arriving wave trains. This quite general statement is fundamentally flawed. In contrast to seismic reflec7on techniques, where the REFLECTED phases at a nega7ve impedance contrast lead indeed to a polarity reversal, this does not apply to refracted waves. For near offsets, the first breaks are formed by the direct wave travelling through the uppermost layer. In the absence of an LVL and the presence of a faster layer at some depth, the first arriving waves at larger offsets are the refracted phases travelling along the upper interface of the high-velocity layer (or it is a diving wave, when the ver7cal velocity changes are more gradual). In the presence of a LVL and the presence of an underlying high-velocity layer, the direct wave con7nues to be the first break (with the same polarity), and the appearance of the refracted phase (with the same polarity) is simply delayed. What may happen is that in the presence of an LVL the amplitude of the first arriving direct wave is reduced and may be difficult to recognize. The authors can confirm all these statements themselves by performing syntheic modeling with horizontal layers. The observation of the authors in the observed and modelled data is likely the result of complicated interreference patterns, caused by the pronounced topography of the various interfaces and the velocity varia7ons. S7ll, the really first arriving waves should not show a polarity reversal, but their amplitudes are probably so small, such that they can no longer be identified.

It might be well possible that the APPARENT polarity reversals in the observed and modelled seismograms, shown in the paper, may be influenced by a LVL, but the general statement that the presence of an apparent polarity reversal is an indica7on of the presence of an LVL is not true, because apparent polarity reversal may be caused by other features of the subsurface (e.g., undula7ons of interfaces and pronounced velocity heterogenei7es).

*Dear Professor Maurer, we appreciate and respect your opinion and we apologise for what we believe to be a fundamental misunderstanding. We are well aware that only reflected waves exhibit polarity reversal in the case of negative contrast in depth, and we are not claiming that head waves exhibit the same polarity reversal phenomenon. However, critically refracted head-waves polarity reversal is a well-known and observed phenomenon associated with the interaction of the wave with a change in medium. We believe, as you correctly suggested, that*

*we are observing just a phase shift due to a complex interaction of the wave delay in the slower layer, which produces the observed phase shift. We already have a constructive comment from the editor on this, and we have already run flat simple modelling with no topography to demonstrate how the LVL generates inversion polarity head wave arrivals, at least as detected by vertical geophones (see Fig.1, which is the same model as in the manuscript but with flat layers and no topography). In our case, we know the composition of the subsurface from the borehole stratigraphy recorded in 1990 and in the summer of 2020, and we have simply demonstrated how the presence of the LVL considerably complicates the shot gather.*

[Figure]

*Fig.1 Synthetic model as in in Fig. 3 of the paper (LVL in purple), but with flat layers showing reversal polarity.*

*On the other hand, we do not intend to state that the polarity reversal in the shot gather implies the presence of an LVL (our statements on this are quite cautious/conservative). This would be a general and fundamentally incorrect statement, as you rightly point out, and far from our original intention. In our experience, the polarity reversal of the head waves has been observed in several papers in different environments and in different rock-glacier datasets. We must take into account this experimental evidence, even if 'apparent' and ascribed to interferences and the limited use of vertical geophones (e.g. horizontal motion does not show polarity reversal in synthetics and real dataset). Nor can it be attributed to incorrect phase picking, as the thickness obtained is consistent with other independent information such as ERT or boreholes. An in-depth study of the complex critically refracted head-wave interaction with LVL is beyond our scope, which is limited to warn RST users in periglacial environments. As you can easily verify from a number of studies, published also in this Journal, SRT data polarity is practically neglected in periglacial exploration, where only time picking is performed, and a simple two-layer model is provided (mostly to define the active layer thickness). In this short communication, we simply want to point out that this seismic attribute may be a proxy for a more complex subsurface stratigraphy as the presence of LVL, which needs to be explored in more detail with other prospecting techniques (as this goal is beyond the potential of common SRT). The communication has been amended starting from the title, in line with your and the Editor's comments, and we thank you both for your invaluable suggestions. We refrained from exceedingly general statements, emphasising that the SRT polarity reversal deserves more attention in the interpretation of the resulting models. We continue to believe that this is an important message to convey to the cryosphere community to avoid a simplistic interpretation of the subsurface in the common use of reflection seismic tomography in rock glaciers. In the revised paper we included all these points in the discussion.*

*Dear Reviewer 2,*
*Thank you for your time. We present here a response to your comment (your comments in black, our responses in italic red).*
*Jacopo Boaga (on behalf of all authors*

Review of the manuscript egusphere-2023-2774: 'Brief communication: On the potential of seismic polarity reversal to detect a thin low-velocity layer above a high-velocity layer in ice-rich rock glaciers'. Co-authored by Jacopo Boaga, Mirko Pavoni, Alexander Bast, Samuel Weber. Submitted to the Special Issue: 'Emerging geophysical methods for permafrost investigations: recent advances in permafrost detecting, characterizing, and monitoring'.

The manuscript presents interpreted field geophysical data (hammer seismic and electrical resistivity) along with numerical simulations to address the possible characterization of a low-velocity layer (LVL) indicated by a 'polarity reversal' on the shot-gather. I believe the authors have perfectly suitable experimental datasets and tools to provide this special issue with a very interesting contribution. However, although labeled as a 'brief communication', the presentation is insufficient to judge the quality and relevance of the results. Several moderate but important comments and major gaps need addressing:

*We thank Reviewer 2 for her/his constructive comments, which can considerably improve our manuscript. For the sake of clarity, we provide a point-by-point response here.*

- Just as the authors take the time to explain what an impedance contrast is, they should propose a simple figure illustrating the structure of a rock glacier and its main characteristics, more particularly linked to the anticipated contrast between electrical resistivity and the seismic velocity of the pressure waves (VP);

*Rev2 is correct and we agree, but TC Brief communications format allows only 3 figures in total, not exceeding 4 pages and does not admit appendix or extra material. We totally modified our figures (as suggested), inserting the RG model as derived from the geophysical information in new panels.*

- -In the same spirit, the author should present, on this first figure, the anticipated LVL and the associated ranges of thicknesses and VP contrasts. The authors should homogenize, by the way, the way in which they mention the LVL throughout the text (sometimes it says 'low-velocity layer (LVL)' and sometimes it's 'low-velocity layer' only)... I think the authors should give the abbreviation the first time and then stick to it;

*Rev2 is right, we homogenized the LVL terms in the manuscript.*

- -The last part of the introduction is not clearly written and is fairly repetitive. It needs to be reorganized, simplified and brought into line with the following sections;

*We modified the introduction according to your suggestion.*

- -The two sites presented are very interesting case studies. However, the Schafberg site is better described than the Flüelapass site. It's important to distinguish between the two (the descriptions need to be better balanced). For each site, the authors have to give more details about the seismic acquisition. The number of shots and their locations should be given as well as the sampling parameters;

*Rev2 is right and we added further information about the Fluela site. The Schafberg site was better described because we have the borehole information that we lack at Fluela. That's why we can trust better in the Schafberg RG model, and we adopted that model for the synthetical simulation. All the seismic acquisition parameters are now inserted as correctly suggested.*

- -As for the interpreted VP and electrical resistivity models, they should be accompanied by more details about the inversion process they involve (information on parameterization, regularization and convergence criteria). As the authors mention that the thin layer is not detectable/recoverable on these models, there should be a discussion about the resolutions of the methods (as well as their depths of investigation). I understand that this is presented in detail (at least for the Schafberg site) in a recent publication (Pavoni et al., 2023a), but I think the gist should be given in the present contribution;

*We inserted more details of the inversion ERT process in the revised manuscript. Resolution limits will be cited, even if ERT resolution capabilities depend on several parameters and a detailed discussion would go beyond the intention of this communication (taking too much space).*

- *-The `brief communication' format requires conciseness, I suppose, so I would recommend that the authors present only one type of style for the seismograms (the 'wiggle' mode is sufficient to see the polarity inversion). In fact, I would suggest changing the logic between Figures 1 and 2 and presenting the models interpreted by SRT (& ERT) on one figure, and the seismic data from which they are derived on the other; As far as the seismograms are concerned, since the models do not depend solely on an end-of-line shot, I would recommend showing at least 3 shot-gathers per line: each end-of-line shot (forward and reverse) and the middle one. This would make it possible to see whether the polarity reversal occurs all along the line (or at least everywhere where the LVL is supposed to exist). I would recommend that the authors separate the figures from the numerical part in the same way. With the models in one figure and the synthetic data in another;*

*We intend to keep this paper as a brief 'alert' communication for SRT users in RG environments and, unfortunately, we can prepare a maximum of 3 figures in total. However, we tried to*

*satisfy your comments by preparing 3 panels in wiggle mode only, for lateral and central shots over the frozen layer. Synthetic modelling must fit into 1 figure only, so we intend to keep the original version (but in wiggle mode as real dataset and as you suggested), which shows the model with and without LVL.*

- *-Regarding the models, as it is full-waveform modelling, I would recommend the authors to provide readers with every parameters (density which is important for impedance and shear-wave velocity (VS));*

*All the parameters are now added according to your suggestion.*

- *-It is also important to provide readers with every modelling parameters, thus making it possible to reproduce the numerical experiment (source parameters, spatial and temporal discretization, boundary conditions etc.). As for the source, does it align well with the experimental one? Did the authors compare their frequency spectra ?*

*We thank Rev2 for this observation, to simulate our sledge-hammer shots realistically we adopted a Ricker wavelet source centred at 60 Hz. This information is now added to the text as all the parameters of the simulation.*

- -Why isn't the Flüelapass case presented/tested in the same way?

*ERT and seismic were collected in the same way and we will add further information about this site. In the Flüelapass case we do not have direct borehole information, so the presence of LVL is only hypothesized. Sensors spacing was different because the active layer at Flüelapass was supposed thinner.*

- -In the same way as for the real data, is it possible to have at least 3 shots per line to see if the LVLs generate a polarity reversal as well, depending on the position (in relation to the surface topography, the shape of the interface, etc.)?

*We are again limited by the number of admitted figures. This would mean that the LVL model would need 4 panels, plus the 4 panels for the model without LVL. In our opinion this would make the single figure too dense and with scarce readability. On the contrary our aim is just to show as the presence of LVL may induce interferences that generates phase inversion. Of course, different topography and thickness may generate other results, however, a multi-parameters test (e.g. changing velocities, LVL thickness, topography, etc.) is beyond the scope of our brief communication and may deserves an extended paper in the next future.*

- -Finally, the detailed analyses of surface waves proposed in the perspectives seem very interesting. I actually did wonder when I asked about the VS model that was used for the simulation. Does it also involve an LVL? How does this fit into the general model for such rock glaciers? It would be interesting to add these hypotheses to the first figure I asked for earlier. If a VS LVL layer is present in the model (and/or in the real world), the dispersion of surface waves should be

influenced. Have the authors tried to calculate the shotgather's dispersion/fk spectra with and without this polarity inversion?

*Rev2 is right and we are aware of the SW potential in RGs. Vs parameter of synthetic data is now inserted. Unfortunately, the real surveys were conducted with refraction aim adopting high frequency geophones. These act as physical filter and do not allow to collect low frequency SW ground roll, which is positive for clear picking of head-waves first arrival but prevents SW analysis. We have recently collected data with 4.5 Hz sensors specifically to study SW in RGs with promising results; we intend to prepare soon another work about SW dispersion in these environments.*

Once again, I believe that the authors have the experimental data and tools perfectly suited to making a very interesting contribution to this special issue. I hope that these few comments and the moderate revisions asked will help them. I look forward to reading a revised version of their manuscript.

*The communication was amended in line with your comments and those of Prof. Maurer (Reviewer 1) and from the Editor, and we thank you all for your valuable suggestions. We continue to believe that this is an important message to convey to the cryosphere community to avoid a simplistic interpretation of the subsurface in the common use of reflection seismic tomography in rock glaciers.*

---

## Author Response (AR3)

**EGUSPHERE-2023-2774-R4**

**Reply to Editor**

*Dear Editor,*
*We present here a response to your comment (your comments in black, our responses in italic red).*
*Jacopo Boaga (on behalf of all authors)*

Dear authors,

thank you very much for submitting a revised version of your manuscript. As you can see from the reviewer's comments, they suggest to add some more detail, particularly on the data and modelling components of your paper. I do agree with the reviewer in that providing more detail will help to better judge the results and modelling steps, but will also make your research more accessible and reproducible. I do understand that a short communication is restricted in space, and one suggestion would be to create a data repository, that not only holds the data, but also the modelling code, and some additional details on the modelling parameters.
Thank you very much again for addressing the reviewers comments.

All the best,
Sebastian Uhlemann

*Dear Dr Uhlemann,*
*we have carefully considered all the suggestions from you and Rev2. We have added details of the SRT and ERT inversions (more specs have been explained in the published references), and we have added information on modelling. As for the source, we have specified that the commonly used Ricker wavelet centred at 50 Hz fits our experimental shot (as you can verify in Figure R1 here below). Unfortunately, brief communications do not admit supplementary materials, but as you suggest a repository (github) with the data and detailed specifications will be created to help results reproducibility. We believe that going into further processing details (which can be found in the reference) goes over the intentions of the TC short communications, also considering that TC policy asks brief comm. to be 'timely' and we are under revision for more than 6 months. We would appreciate your decision on this, whatever it may be. Thank you for your time.*

*Dear Reviewer 2,*
*Thank you for your time. We present here a response to your comments (your comments in black, our responses in italic red).*
*Jacopo Boaga (on behalf of all authors)*

The authors have partially modified their figures as requested. I thank them for these efforts. However, it seems important to find more information on the assumption of LVL existence in such a structure and its geometry (anticipated thickness and ranges of mechanical properties, their correspondence in terms of resistivity). In addition, the authors should give a minimum of details on the inversion parameters and criteria used for their ERT and SRT models. The length of the paper is indeed strict, but I think part of it can be shortened to give space to essential parameters (information on parameterization, regularization and convergence criteria). As far as modeling is concerned, the authors provide some information on the source, but several questions remain unanswered: "It is also important to provide readers with every modelling parameters, thus making it possible to reproduce the numerical experiment (source parameters, spatial and temporal discretization, boundary conditions etc.). As for the source, does it align well with the experimental one? Did the authors compare their frequency spectra ?". I think the authors have the experimental data and tools perfectly suited to make a very interesting contribution to this special issue. I hope these few comments and the minor revisions requested will help them. I look forward to reading a revised version of their manuscript.

*We thank Reviewer 2 for her/his constructive comments, which considerably improved our manuscript. In particular, your very interesting suggestion to show lateral and central shots highlights as in the right part of the Fluelapass site, that miss the LVL, doesn't present the phase reversal. This strengthened our hypothesis. The communication was amended in line with your comments and from the Editor (see highlighted version), and we thank you both for your valuable suggestions. Some sentences were deleted to fit the pages limit, and we now introduced general information about common LVL in RGs thickness, electrical and mechanical properties (ln 43-48). We then added information about SRT and ERT inversion (ln 75-85 related to the cited published reference), whith data errors (previously in the figures captions). Unfortunately, no supporting files are allowed by TC, but following the suggestion of the Editor we will add all the info attached here (see appendix 1) in an available repository with the data (and this appendix too will be also public as the paper discussion). Modelling parameters are now provided in lines 116-120, specifying we adopted adaptive mesh with 10 elements per wavelength and external absorbing boundary. The forward solution time sampling is not a chosen parameter in Salvus, since it is automatically fixed by the code for the algorithm stability (in our case was 198.01 KHz, for the details we refer to Salvus documentation as in Afanasiev et al., 2019). Source parameters of the modelling are now explained in par.3. We adopted a commonly used Ricker wavelet centred at 50 Hz, since it fits our experimental field*

*shot. The sources have in fact a power range of 10-90 Hz with average central peak around 40-50 Hz (see an example in the figure R1 here below).*

[Figure]

*Fig. R1 a) experimental recordings of 1 shot at Schafberg site with 20kg sledgehammer, time and frequency domain; b) Ricker wavelet adopted for synthetic modelling centred at 50 Hz.*

*We thank Rev2 and we believe her/his suggestions considerably improved our work. We continue to believe that this short contribution is an important message to convey to the cryosphere community to avoid a simplistic interpretation of the subsurface in the common use of reflection seismic tomography in rock glaciers.*

**Appendix 1 Technical info for SRT / ERT processing**
* * *
***Flüelapass site***
- ➢ *ERT Acquisition*

*Syscal Pro- device, 48 channels, 2 m spacing, Dipole-Dipole skip 0-3, stacking range 3-6 (5% standard deviation threshold), and direct and reciprocal measurements.*

- ➢ *ERT inversion modelling ResIPy*

*Filtering*
- $\rho_a < 0$
- $stacking\ error < 5\%$
- $reciprocal\ error < 10\%$  (1050/1901)

*Inversion modelling*
- *Inversion type: regularized inversion with linear filtering;*
- *Regularization mode: normal regularization;*
- *Data type: logarithmic;*
- *Expected data error: 10% (a_wgt = 0.01, b_wgt = 0.10);*
- *Flux type: 3D;*
- *Weights update: routine based on Morelli and LaBrecque (1996);*
- *smoothing factor: normal isotropic regularisation (= 1);*
- *Iteration: 2;*
- *Final RMS misfit: 1;*

*Expected data error evaluated with the reciprocal check. We defined a boundary threshold for the reciprocal error that allowed for a reliable quality of the measured apparent resistivities but at the same time a homogeneous distribution of measured points in the pseudo-section.*
*We applied an isotropic smoothing since we were interested in highlighting both lateral and vertical variations of resistivity.*

- ➢ *SRT acquisition*

*Geode seismographs, 48 channels, 100 Hz geophones, 2 m geophones spacing, 4 meters shots spacing, 2 shots in each position, 20 kg hammer as seismic source.*

- ➢ *SRT inversion modelling Pygimli*

*Inversion modelling:*
- *Picking error: 2 ms*
- *smoothing factor: normal isotropic regularisation (= 1);*
- *Regularization factor λ: 150;*
- *Starting model: gradient model 300-3000 m/s;*
- *Iteration: 4;*
- *Abort criteria reached: dPhi = 1.26 (< 2.0%);*
- *rms/rrms(data, Response) = = 0.00489155/14.2038%;*
- *chi^2(data, Response, error, log) = 5.48181;*

*Picking error: we evaluated the data uncertainty by performing a repeated picking of P-wave first arrivals for several shot gathered, calculating this way a representative standard deviation of 2 ms.*
*Regularization factor: we chose λ values using the L-curve analysis.*
*We applied an isotropic smoothing since we were interested in highlighting both lateral and vertical variations of Vp.*
* * *
***Schafberg site***
- ➢ *ERT Acquisition*

*Syscal Pro- device, 48 channels, 3 m spacing, Dipole-Dipole skip 0-3, stacking range 3-6 (5% standard deviation threshold), and direct and reciprocal measurements.*

- ➢ *ERT inversion modelling ResIPy*

*Filtering*
- $\rho_a < 0$
- $stacking\ error < 5\%$
- $reciprocal\ error < 20\%$ (saved 1029/1901)

*Inversion modelling*
- *Inversion type: regularized inversion with linear filtering;*
- *Regularization mode: normal regularization;*
- *Data type: logarithmic;*
- *Expected data error: 20% (a_wgt = 0.01, b_wgt = 0.20);*
- *Flux type: 3D;*
- *Weights update: routine based on Morelli and LaBrecque (1996);*
- *smoothing factor: normal isotropic regularisation (= 1);*
- *Iteration: 2;*
- *Final RMS misfit: 1.17*

*Expected data error evaluated with the reciprocal check. We defined a boundary threshold for the reciprocal error that allowed for a reliable quality of the measured apparent resistivities but at the same time a homogeneous distribution of measured points in the pseudo-section.*
*We applied an isotropic smoothing since we were interested in highlighting both lateral and vertical variations of resistivity.*

> *SRT acquisition*

*Geode seismographs, 48 channels, 100 Hz geophones, 3 m geophones spacing, 4 meters shots spacing, 2 shots in each position, 20 kg hammer as seismic source.*

> *SRT inversion modelling Pygimli*

*Inversion modelling:*
- *Picking error: 2 ms*
- *smoothing factor: normal isotropic regularisation (= 1);*
- *Regularization factor λ: 200;*
- *Starting model: gradient model 500-5000 m/s;*
- *Iteration: 4;*
- *Abort criteria reached: dPhi = 0.42 (< 2.0%)*
- *rms/rrms(data, Response) = 0.00309603/17.893%*
- *chi^2(data, Response, error, log) = 2.39635;*

*Picking error: we evaluated the data uncertainty by performing a repeated picking of P-wave first arrivals for several shot gathered, calculating this way a representative standard deviation of 2 ms.*
*Regularization factor: we chose λ values using the L-curve analysis.*
*We applied an isotropic smoothing since we were interested in highlighting both lateral and vertical variations of Vp.*